# Smart-Card Technology for the Dental Management of Medically Complex Patients

**DOI:** 10.3390/healthcare10112314

**Published:** 2022-11-18

**Authors:** Mohammed Alshehri, Abdullah Alamri, Mohammed Alghamdi, Rakan Nazer, Omar Kujan

**Affiliations:** 1Dental Department, King Khalid University Hospital, King Saud University, Riyadh 11545, Saudi Arabia; 2Cardiac Sciences Department, College of Medicine, King Saud University, Riyadh 11545, Saudi Arabia; 3UWA Dental School, The University of Western Australia, Nedlands, WA 6009, Australia

**Keywords:** medical record linkage dentistry, patient care, medically complex, medical records, medical informatics

## Abstract

Smart-card technology is believed to help healthcare industries in several ways, since it minimizes risks and medical errors, enables accurate patient identification, reduces administrative costs, improves efficiency, and facilitates prompt delivery of care to patients. The present study aims to highlight the adoption of a newly designed dental smart card for medically complex patients. The present smart card is an advance in patient identification, using a quick-response (QR) code to automatically report or receive certain types of responses from patients or physicians once illuminated by signals from QR readers. Further, the card provides general information about the patient’s condition and physical details. The card is pocket sized and can be carried easily by the patient anywhere, alongside a digital copy of the card.

## 1. Introduction

Patient identification plays a significant and vital role in managing patient safety in any health care system. Existing evidence supports a systematic approach, with instructions to be followed in association with patient identification to avert errors and undesirable consequences in health care organizations at each stage of patient care [1]. A joint collaboration between the World Health Organization (WHO) and Joint Commission International in 2007 advocated specific and important patient safety factors to prevent adverse events and errors in a healthcare system [2]. These solutions were based on interventions, systems, and projects targeting patients’ overall safety and reducing harm, including correct identification of patients as an important tool that can attenuate patient harm [3].

Nevertheless, in many healthcare organizations, patient identification is performed manually with the help of healthcare staff. However, this technique can result in anthropogenic errors; therefore, electronic methods serve as an effective substitute. Studies have shown that identification errors can impact patients, as well as healthcare professionals, by leading to severe problems associated with medical procedures. These problems can include errors in the administration of medications, blood transfusions, and minor or major surgery [4]. These errors can be minimized using automated electronic systems for patient identification [4,5]. Automated patient identification systems have several advantages: they are quick and reliable in identifying patients and facilitate easy accessibility and arrangement of patient medical records. Hence, they maximize the safety and security of patient identification, delivering correct clinical information quickly and effectively [6]. In addition, they facilitate the establishment of national registries that can ultimately improve the delivery of health services and enhance the efficiency of the health care system [7,8]. They have, therefore, gained immense popularity in medical and dental practices.

This paper aims to describe the adoption of smart cards using quick-response (QR) codes for the dental management of medically complex patients. It also summarizes the clinical challenges and impact of this technology in providing the best standard of dental care. Hence, this paper will highlight the significance of the dental management of medically compromised patients, the use of QR-based technology in healthcare, and novel technologies that can enhance treatment outcomes. One such emerging technology proposed in this paper is the introduction of a smart card that will contribute substantially to the timely recognition and management of adverse events in dental settings. This smart card will work as a safety signal and identify the characteristics of the patient and their associated diseases to provide a basis for altering the treatment plan. A search strategy based on the key words “Medical records, QR, systemic conditions, medically complex, and dentistry” was performed in Medline and Scopus. Papers published between 1945 and June 2022 were scrutinized. Only articles that describe the QR terminology and their applications in dentistry were selected and discussed in this review.

## 2. The Impact of Patient Records on the Dental Management of Medically Complex Patients

The dental management of medically complex patients requires an understanding of their diseases or conditions. There are numerous diseases that can interfere with dental planning and treatment, including cardiovascular diseases such as acute coronary syndrome (ACS), which is characterized by clinical symptoms of severe chest pain and pressure on the chest caused by a compromised blood supply to the myocardium. The underlying condition of ischemic heart disease is the primary cause of ACS. It can present with symptoms at rest or while engaging in physical activity. ACS challenges the clinician at each treatment step in terms of diagnosis, risk stratification, therapeutic decision-making, and monitoring response to therapy. The underlying condition of ischemic heart disease is the primary cause of ACS. It can present with symptoms at rest or while engaging in physical activity. Therefore, it becomes essential for dental practitioners to perform risk assessments before initiating dental treatment. Such medically compromised patients must undergo complete medical investigations and referrals by their doctors to evaluate their complications, treatment protocols, and emergency care [9].

Dental practitioners should be well informed about the complete medical history of all patients, as well as possible treatment modalities and medications for medically compromised patients. Additionally, dentists should be aware of all types of medical emergencies to identify and implement the necessary emergency care protocol effectively in such situations [9].

As mentioned earlier, medical history is recorded manually (a natural medium being paper) since it is easy and straightforward, is cost-effective, and is a durable approach. Medical information can be written quickly on paper and exchanged among healthcare professionals involved in a particular patient’s treatment. However, it becomes problematic when professionals in different healthcare settings are involved. The decentralized nature of the healthcare system creates several problems. For example, a patient with cardiovascular disease might visit different practitioners to receive treatment for their oral problems, cardiology issues, and any other primary care concern, which could impede the flow of complete medical information. As a result, patient health records become fragmented, creating issues in delivering quality care to patients [10]. In addition, administrative costs add to financial burdens; hence, practical measures that could alleviate these costs are required [11].

## 3. Emerging Technologies in Medical and Dental Practices

Over the past few years, automated patient identification systems have revolutionized medical and dental practices. With the advent of technology, healthcare companies have designed advanced models for such systems. One such method uses graphical one-dimensional (1D) codes [12], also known as barcodes, commonly printed on labels. This technology has also been realized as radio frequency identification tags [13]. It is used mainly by healthcare centers; however, technological differences can also exist. Another method, barcode technology, can be modified into two-dimensional (2D) graphical codes [4]. These novel 2D versions of barcodes are believed to improve existing one-dimensional (1D) barcodes, although they can be printed and generated like 1D codes, and their better design helps in the storage of large amounts of information and data. Additionally, they have the key characteristics of being able to identify errors and promote error correction [14]. Numerous options with 2D codes are available that allow for the identification of patients, but QR-code [15] technology has gained significant popularity in the healthcare industry due to its increased general usage and the increasing numbers of applications relying on QR codes [16].

Denso Wave developed QR codes in 1994. Denso Wave was a division of Toyota that used QR codes to track components of cars during manufacturing and distribution [17]. The patent for QR codes is owned by Denso Wave; however, this technology is available free of cost worldwide. The use of these black-and-white pixelated squares has been increasing rapidly. Owing to their ability to store more significant information (in one-tenth of the space) than conventional barcodes, their high speed, and their scanning properties, they have become one of the most reliable tools for enhancing patient safety and security.

## 4. QR-Based Technology in Healthcare

Due to their versatility, QR codes have been used in all types of healthcare settings. For example, in maxillofacial radiology, they help to store patient case histories [18]; older patients can keep better track of their use of medications for increased safety [19]; and, following orthopedic cast application, QR codes can provide instructions to patients [20]. They also play a valuable role in healthcare education by increasing students’ access to visual learning materials [21].

In medicine, effective monitoring of patient health is essential. In several situations, due to the unavailability of proper medical records, it becomes challenging for medical staff to provide proper treatment for critically ill patients who are comatose and placed in the emergency department. Physicians’ inability to access patients’ past medical histories can impair the treatment planning, including the surgical protocol, required to proceed further. Hence, QR technology will provide a health-monitoring system for patients and healthcare professionals, allowing for easy accessibility of medical information anywhere at any time, irrespective of disparities in healthcare settings [22]. Physicians can quickly access medical records and evaluate a patient’s health status. This helps them to deliver quality care in emergency situations, thereby enhancing patient safety. This technology provides a complete and easily acquired record file of any patient. Furthermore, electronic-based patient records accessible via QR codes allow for the sharing of medical information within the group of doctors involved in treatment, ensuring the exchange of full records (including allergies, past medical history, and medications) when required [23].

QR-code technology has numerous advantages for patient identification: this method is user-friendly and provides ease of access since it does not require any unique tags (such as radio frequency identification tags) for identification. QR codes can be generated easily on any surface, such as paper or plastic labels; hence, no specific equipment is needed to print them, and a simple printer is sufficient [4]. QR codes are widely used because smartphones are now being used in all aspects of life, which makes reading and decoding these codes quite simple and popular compared with more complex systems. Compared with radio frequency identification-based systems, reading QR codes is only possible when one is in close proximity, making it impossible to read a code that is not identifiable or has errors. This fact makes QR-code reading unambiguous compared with other technologies currently being used in healthcare organizations. As mentioned earlier, QR-code technology is more effective and superior to other systems in terms of its higher storage capacity, lower cost, user-friendliness, accessibility, technical simplicity, approachability, and electronic learning via camera-equipped smartphones. All of these features make QR-based technology a promising system that can be widely implemented for patient identification purposes, even in developing countries that struggle with limited resources and lack health care support [4].

## 5. Use of QR-Based Technology in Dentistry

### 5.1. Denture Labeling

Over the past few decades, QR-based technology has been used in dentistry. Applications related to denture-labelling barcode generators are used by dental practitioners, and fast scans are performed by smartphones. Since QR matrix barcodes can store a large amount of data, they are an ideal system for denture labelling [24]. This method allows for the placement of the patient’s identification within the denture, making it simple for the technician, as well as the dentist, to minimize the risks of denture loss. Interestingly, this process of denture labelling is also implemented in the forensic sciences, in which it helps in correct identification during catastrophic events [25].

### 5.2. Inventory Control

Another critical role of QR codes in dentistry is their use in inventory control inside health organizations. They are essential in managing dental materials, acting as a physical store in which adequate tracking of material in use is performed. Hence, they enable materials to be appropriately categorized and selected and their use to be analyzed statistically [26].

### 5.3. Drug Prescription

A common medical error encountered widely in the healthcare industry is incorrect drug prescription. A US-based study noted that a large portion of the population (approximately 1.5 million people) is affected by incorrect drug prescriptions annually, costing $3.5 billion on associated treatment. Therefore, the use of QR codes is being popularized because it will minimize such errors by creating an accurate health record, one in which all of the risk-associated factors for medically complex patients can be tracked. Moreover, if a physician is well informed about a patient’s health condition, he or she can communicate the information to the patient and family members. Doing so will further prevent the occurrence of medical errors [22].

### 5.4. Dental Education

QR-code technology helps students to gain knowledge quickly by promoting access to a wide variety of scientific materials via the internet. For example, e-learning can be implemented in dentistry by scanning any QR-tagged dental instrument required for any procedure, and the associated resource or training sessions will be made available. It also helps students to conduct online surveys and access feedback and evaluation forms at any dental institution, and it provides direct access to presentations and lecture notes for dental students [27].

### 5.5. Dental Practice Management

QR-based technology can prove beneficial in dental practice in two ways. First, it can help to build mutual trust and connection between the dentist and the patient. Second, it can provide feedback associated with patient satisfaction, physician communication skills, and quality of care delivered by any hospital or individual practitioner in a dental setting. Education facilitates the exchange of dental knowledge, videos, and research materials between practitioners [27]. This ease of use will help dental practitioners to improve and deliver high-quality treatment to all patients.

Broadly speaking, QR technology is vital to medical and dental care management. The key advantages of QR technology [22] are that it

reduces the risk of medical errors;allows for correct and accurate patient identification;improves patient care and management;positively impacts patient survival rate; andpromotes dental education.

## 6. Adoption of a Newly Designed Dental Management Card

### 6.1. An Approach to Improving Patient Care

The medical card is a card that will provide a record of a patient’s personal health information during routine medical examinations and emergencies. In the case of a medical emergency, it is extremely important for medical staff to have immediate and fast access to the patient’s medical history. A medical history of conditions such as chronic diseases, allergies and blood diseases must be known before starting any treatment to prevent undesirable harm or damage to the patient due to an existing medical condition. Usually, this kind of information is difficult to gain without the help of the patient, especially in case of medical emergencies. To avoid this problem, new methods have been introduced, such as medallions and bracelets that can be worn by patients with particular medical conditions, such as diabetes, cardiac disease, hemophilia or antibiotic allergies. However, there is a need to provide a more adequate and comprehensive means of describing a patient’s medical records on a device which can be easily carried by all patients. None of the previous literature has revealed a device that properly displays a patient’s significant medical history in a way that is reasonably easy to read and use by medical staff and which is sufficiently adaptable to production.

Based on the foregoing systems that help in the medical treatment of the patient, it is expected that the patient’s smart card will become a very important piece of equipment. Therefore, there is a continued need for cost-effective technologies to enhance the functionality and usability of smart cards and related systems. Of course, medical information is highly sensitive and has to be treated with a high level of security. At the same time, patients can benefit from a high level of built-in functionality in a single, secure device. It is worth noting that the current smart card represents a new and enhanced approach to personal health, as it includes several advanced technologies. An essential feature of the patient smart card is a QR code that, when subjected to a signal from a QR code reader, will automatically present certain information, such as an authorized doctor’s recommendations or an update to the patient’s record. In this paper, we present a smart card designed to resolve and surmount existing technical difficulties and shortcomings.

### 6.2. Dental Card Components

A wallet-sized medical smart card that can be carried by a person, comprising

a card with two sides, laminated with a clear plastic material;a portion on the front side of the card for identification data, including data identifying the person carrying the card;a portion including patient doctor identification data;a portion for an emergency data code containing a list of health conditions that should be listed with a symbol indicating the condition, treatment protocols and precautionary measures listed per the patient’s requirements; anda QR code printed on one side that allows for updated information transfer between the patient and doctor without the need for a physical checkup.

### 6.3. Implementation

This smart card can be made into a multipurpose personal health card, where patients’ information is securely held in cloud storage. Figure 1 is an example of a wallet-sized personal health card design. The card includes an embossed account number, hologram, and other marks carried on conventional credit and debit cards. Of course, not all of these features should be included, but additional features can be added if needed for a particular application. The features presented in Figure 1 comprise a powerful tool to implement a greatly improved new paradigm in the management of personal health care and insurance.

The health card will typically include the patient’s name, file number, medical condition, medical center contact details, the patient’s physician’s name and QR code. On the back side of the card, multiple protocols and precautions will be listed that are customized according to the patient’s specific requirements, Figure 2. The QR code will allow patients to see the precautions and show them to their dentists without editing and will provide access for treating physicians to see, update, and edit recommendations and laboratory results. An online platform can be developed, and all data can be held in cloud storage; the patients and doctors can access the data through a web browser or phone application. Licensed and verified doctors can subscribe to this service to create an account and get access to the platform to be able to add new patients and edit previous patients’ data, as shown in Figure 3. While a specific card design has been presented and described, many variations are possible, and additional features could be added.

## 7. Discussion

The previous literature has reported several limitations associated with a lack of effective implementation of QR-based technology: a system that is inadequately aligned with smart cards, a lack of technical support, a complex card framework, and the financial burden on health care organizations to ensure adequate infrastructure support of the technology. This study considered these obstacles and found that the smart card can be designed to overcome all of these challenges. The card is simple to use, allows easy access to the patient’s medical information in emergency situations, and provides all data about the past and present health status of the patient, irrespective of his or her location. Medical information can be quickly exchanged among treating health professionals, optimizing the treatment outcome during any complicated dental procedure.

This smart card will work as a safety signal and identify the patient’s characteristics and associated medical diseases, based on which the treatment plan can be altered. The scope of medications that might interfere with the dental procedure will be tailored to the patient’s health status. Therefore, this smart card will substantially minimize risks and complications in medically complex patients and provide health care providers with detailed information on the patient’s medical history in emergency conditions. Overall, smart cards will play a significant role in the timely recognition and management of adverse events in dental settings.

The present study has certain limitations. First, the validity of the data collection framework should be clearly developed to implement the card in clinical practice. Second, the extent of patient confidentiality possible with this new tool needs further evaluation. Third, the technologies that can smoothly allow QR-code scanning, such as 4G/5G, edge computing, cloud computing, etc., should be compared to develop a smooth and easy transition for patients.

## 8. Conclusions

The present smart card technology for the dental management of medically complex patients will provide both clinical and financial solutions with a single card. Complete storage of the patient’s unique biometrics on the card will make it simple to use, allowing for rapid identification. Additionally, it can be regularly updated to include the most recent medical information from follow-up visits and laboratory results, and it can be exchanged among the treating dentists and physicians. This novel dental card is a portable health record that will significantly improve treatment and management practices in dentistry. Further randomized clinical trials should be conducted that allow for patient verification by smart card technology. Doing so will contribute to a better understanding of clinical, personal, and financial information and any challenges encountered with the card’s use.

## Figures and Tables

**Figure 1 healthcare-10-02314-f001:**
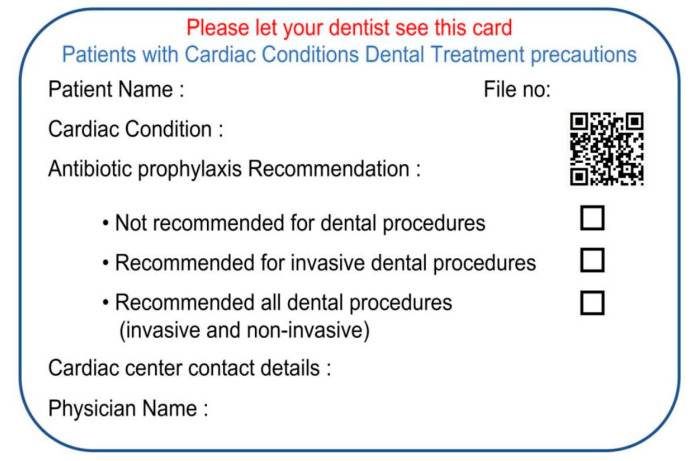
The smart card shows the patient’s data, medical condition, center data, QR code, and physician’s name.

**Figure 2 healthcare-10-02314-f002:**
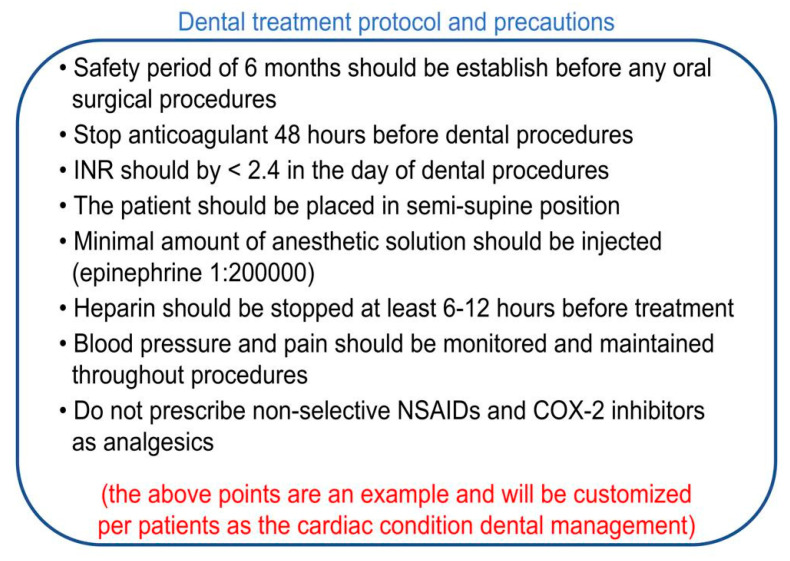
The back side of the smart card shows an example for multiple precautions and protocols that should be followed during dental management of a cardiac patient.

**Figure 3 healthcare-10-02314-f003:**
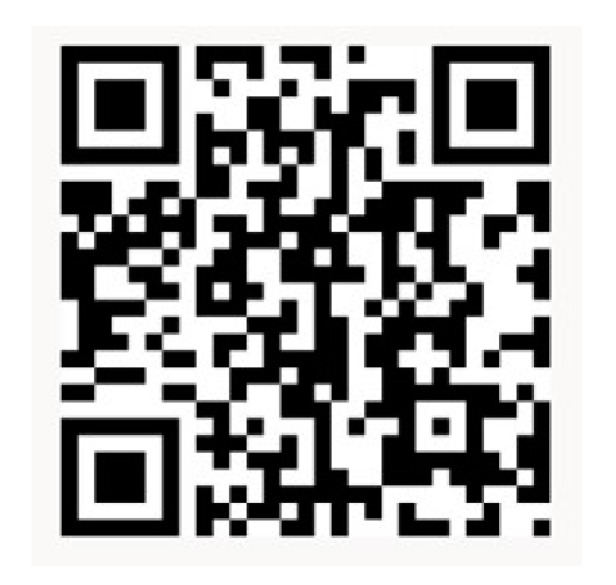
QR code to access patient’s medical records.

## Data Availability

Not applicable.

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
