# Peer review of "Smart-Card Technology for the Dental Management of Medically Complex Patients"

_healthcare, 2022, doi:10.3390/healthcare10112314_

Round 1
Reviewer 1 Report
The study described the use of smart cards using quick-response (QR) codes for dental management for patients with complex medical issues. The researchers identified the impact of patient records on dental management. Then, they described the use of QR-based technology in healthcare in general and in dentistry in particular. After that, they explained the design of the new dental management card. The paper has the following limitations:
There is no implementation or experiment that shows the effectiveness of the new card.
The paper does not discuss the disadvantages of using QR technology in healthcare.
The paper lacks a framework for implementing such technology and integrating it with other systems such as EHR.
Author Response
The study described the use of smart cards using quick-response (QR) codes for dental management for patients with complex medical issues. The researchers identified the impact of patient records on dental management. Then, they described the use of QR-based technology in healthcare in general and in dentistry in particular. After that, they explained the design of the new dental management card. The paper has the following limitations:
There is no implementation or experiment that shows the effectiveness of the new card.
Authors’ response: Thank you. This manuscript is a review and it includes our proposal for an innovative approach. The experimental study is undergoing and will follow this report.
The paper does not discuss the disadvantages of using QR technology in healthcare.
Authors’ response: Thank you. The disadvantages associated with QR use in healthcare have been discussed (page 7).
The paper lacks a framework for implementing such technology and integrating it with other systems such as EHR.
Authors’ response: Thanks. This has also been discussed (page 6).
Reviewer 2 Report
Although the use of technologies in the disease management is a very attractive idea for research, however, the purpose of the study has not been determined. No information about the methods is provided. Also, there are no discussion section.
Author Response
Although the use of technologies in the disease management is a very attractive idea for research, however, the purpose of the study has not been determined. No information about the methods is provided. Also, there are no discussion section.
Authors’ response: Thank you. First, this manuscript is a narrative review, and it doesn’t include the methods or results section. However, the authors have clearly stated the paper’s aim, which is to describe the adoption of smart cards using quick-response (QR) codes for the dental management of medically complex patients. It also summarizes the clinical challenges and impact of this technology in providing the best standard of dental care. The paper proposes using a smart card that will contribute substantially to the timely recognition and management of adverse events in dental settings. This smart card will work as a safety signal and identify the characteristics of the patient and their associated diseases to provide a basis for altering the treatment plan.
Reviewer 3 Report
The author's study aims to demonstrate how a newly developed dental smart card can be used by people with complex medical demands. The patient's physical attributes and general state of health are also described in the card. The patient can simply travel with both the pocket-sized card and a digital version of the card. Their idea appears original and has been technically executed beautifully; as a result, it merits publication. The paper has to be proofread, though, as there are some grammatical mistakes in it. The concepts in the abstract are interesting, perceptive, and relevant to future research. The essay is written exceptionally effectively. Having stated that, the piece can be published.
Author Response
The author's study aims to demonstrate how a newly developed dental smart card can be used by people with complex medical demands. The patient's physical attributes and general state of health are also described in the card. The patient can simply travel with both the pocket-sized card and a digital version of the card. Their idea appears original and has been technically executed beautifully; as a result, it merits publication. The paper has to be proofread, though, as there are some grammatical mistakes in it. The concepts in the abstract are interesting, perceptive, and relevant to future research. The essay is written exceptionally effectively. Having stated that, the piece can be published.
Authors’ response: Thank you for taking the time to review our work. Proofreading has been done to improve the clarity of the manuscript.
Reviewer 4 Report
Dear sir
This research paper is good, but needs more work in terms of experimental test/ and or sample test to know the validity of the research aims and how much match the reality and the possibility to make it applicable.
I suggest the authors to take these notes above then I can reconsider this paper to review it
Good Luck
Author Response
This research paper is good, but needs more work in terms of experimental test/ and or sample test to know the validity of the research aims and how much match the reality and the possibility to make it applicable.
I suggest the authors to take these notes above then I can reconsider this paper to review it
Authors response: Thank you. This manuscript is a review and it includes our proposal for an innovative approach. The experimental study is undergoing and will follow this report.
Round 2
Reviewer 2 Report
Dear authors
Thank you for your detailed response to the comments.
However, you need to explain how to access the desired articles and the criteria for selecting sources. Also mention the databases used and the words and phrases you used to search for the sources.
Author Response
Authors’ response: Thank you for taking the time to review our revision and to provide your appreciated comments. We added a short paragraph outlining the search strategy, search databases and the inclusion criteria .Reviewer 4 Report
Dear Author
This paper is a visionary paper which is good but as long as this paper dose not have experimental study it would be hard to accept it. So I would repeat that experimental test/ and or sample test to know the validity of the research aims and how much match the reality and the possibility to make it applicable is a mandatory terms of acceptance.
All the best
Author Response
Authors’ response: Thank you for taking the time to review our revised manuscript. Though we appreciate your comments, we believe they are irrelevant. You are mandating us to undertake an experiment study while our manuscript is a REVIEW that includes a hypothesis which is very typical. We agree that an experimental study is the only method to either support or refute our hypothesis. Since you firmly won’t accept a hypothesis without proof, I don’t know what science would look like without sharing new ideas and theories!